# gjSOX9 Cloning, Expression, and Comparison with gjSOXs Family Members in *Gekko japonicus*

Xingze Huang [1], Ruonan Zhao [2], Zhiwang Xu [2], Chuyan Fu [1], Lei Xie [2,3], Shuran Li [2,3], Xiaofeng Wang [1,3,*] and Yongpu Zhang [1,2,3,*]

[1] Department of Biotechnology, Life and Environmental Science College, Wenzhou University, Wenzhou 325003, China

[2] Department of Bioscience, Life and Environmental Science College, Wenzhou University, Wenzhou 325003, China

[3] Zhejiang Provincial Key Laboratory of Water Environment and Marine Biological Resources Protection, Wenzhou University, Wenzhou 325003, China

[*] Correspondence: wangxiaofeng@wzu.edu.cn (X.W.); zhangyp@wzu.edu.cn (Y.Z.)

**Abstract:** SOX9 plays a crucial role in the male reproductive system, brain, and kidneys. In this study, we firstly analyzed the complete cDNA sequence and expression patterns for SOX9 from *Gekko japonicus* SOX9 (gjSOX9), carried out bioinformatic analyses of physiochemical properties, structure, and phylogenetic evolution, and compared these with other members of the gjSOX family. The results indicate that gjSOX9 cDNA comprises 1895 bp with a 1482 bp ORF encoding 494aa. gjSOX9 was not only expressed in various adult tissues but also exhibited a special spatiotemporal expression pattern in gonad tissues. gjSOX9 was predicted to be a hydrophilic nucleoprotein with a characteristic HMG-Box harboring a newly identified unique sequence, "YKYQPRRR", only present in SOXE members. Among the 20 SOX9 orthologs, gjSOX9 shares the closest genetic relationships with *Eublepharis macularius* SOX9, *Sphacrodactylus townsendi* SOX9, and *Hemicordylus capensis* SOX9. gjSOX9 and gjSOX10 possessed identical physicochemical properties and subcellular locations and were tightly clustered with gjSOX8 in the SOXE group. Sixteen gjSOX family members were divided into six groups: SOXB, C, D, E, F, and H with gjSOX8, 9, and 10 in SOXE among 150 SOX homologs. Collectively, the available data in this study not only facilitate a deep exploration of the functions and molecular regulation mechanisms of the gjSOX9 and gjSOX families in *G. japonicus* but also contribute to basic research regarding the origin and evolution of SOX9 homologs or even sex-determination mode in reptiles.

**Keywords:** *Gekko japonicus*; gjSOX9 cDNA; expression; bioinformatics analyses; gjSOX family





## 1. Introduction

As transcription factors, all SOX (SRY-like HMG Box) proteins contain a DNA binding domain (a high-mobility group (HMG)) and share a homologous HMG Box with SRY (sex-determining region Y, a decisive factor on the mammalian Y chromosome). The SOXs family consists of groups (A-K) [1–4] which play various roles in cell fate decision, tissue formation, sex determination and differentiation, and morphogenesis [1,5–8].

SOX9 is a master member of the SOXE group of the SOXs family. Previous studies have shown that SOX9 plays dominant roles in male sex determination and testis differentiation in red-eared slider turtle embryos [9]. Another investigation showed that the deletion of 5′ distal enhancer13 (Enh13) from the *SOX9* gene led to complete XY male-to-female sex reversal, indicative of SOX9′s crucial roles in testis development in mice [10]. In teleost fishes, two duplicates of SOX9 (SOX9a and SOX9b) share distinctly different expression patterns and perform different functions, particularly in sex differentiation and reproduction [8,11,12]. SOX9, along with its family members, has been well reviewed for its function and regulation in testis determination, gonad differentiation, and male fertility

maintenance [3,13–15]. In addition, SOX9 plays pivotal roles in brain tissue formation and functional maintenance [16], nephrogenesis in *Alligator sinensis* [17], and chondrocyte and disease occurrence in mammals [18].

The SOX9 transcription factor shares a characteristic HMG-Box region, a highly conserved structure, with its family members. SOX9 regulates gene expression via its DNA binding domain by interacting with other transcription factors or cofactors, post-transcriptional and post-translational modifications [7,13,18], or via nucleocytoplasmic shuttling [7,17].

Genome data provide valuable resources for identifying genes or gene families via genome-wide identification or characterization [19,20]. BLAST analyses indicated that SOX9 orthologs were present in almost all metazoans [21]. Although a number of SOX9 genes have been cloned and analyzed in species such as the mud crab *Scylla paramamosain* [21], *Alligator sinensis* [17], and *Astyanax altiparanae* [22], the gjSOX9 cDNA sequence and expression patterns, and comparisons between gjSOX9 and gjSOXs family members, remain unknown.

*G. japonicus*, an important member of the genus Gekko in the family Gekkonidae, possesses significant features such as tail regeneration, robust smooth climbing abilities, and super-sensitive olfactory and night vision abilities. In particular, its sex determination is complexly co-regulated by genetic and environment factors [23]. To date, there are still different views on whether the sex determination mode of *G. japonicus* is XX/XY, ZZ/ZW, or temperature-dependent [24–26]. This is perhaps mainly due to the indistinguishable chromosome number and structure. Furthermore, *G. japonicus* produces a high ratio of female individuals at incubation temperatures of 24 °C or 32 °C, while the sex ratio at 28 °C is almost 1:1, so it is difficult to determine whether sex reversal occurs among individual offspring, which brings tremendous challenges when seeking sex DNA markers or investigating the mode of sex determination. Genome data show that several core sex-related genes, such as *SOX9* and *DMRT1*, are present in *G. japonicus* [27]. Exploring the cDNA sequences, expression profiles, and phylogenetic evolutions of these sex-related genes to investigate their function would help to more deeply understand sex determination and development mechanisms in *G. japonicus*.

In this study, the gjSOX9 cDNA sequence and the gjSOX9 expression pattern were determined. Additionally, the properties and conserved regions of gjSOX9 were identified, and comparisons were made with its orthologs, gjSOX family members, and family homologs among species. The available data provide valuable clues to better investigate the function and evolution of SOX9 and its family members in *G. japonicus*.

## 2. Materials and Methods

### 2.1. Samples and Tissues Collected

*G. japonicus* adult individuals (at least 3 years old) were sampled in Wenzhou, China (120°41′27.30″ E, 27°55′17.73″ N), and a group of 5 female and 5 male individuals were raised in each cage in the lab for possible mating and spawning. *G. japonicus* babies were generated from hatched fertilized eggs in 28 °C incubators (the temperature for producing a 1:1 sex ratio in offspring) and raised with adequate water and food in a suitable environment.

Adult testis tissues were isolated for gjSOX9 cDNA cloning. gjSOX9 expression patterns were determined using adult tissues (gonad, liver, kidney, brain, heart, and muscle), adult gonad tissues during the reproductive period, May (generally from April to August), and the non-reproductive period, November (generally from September to March), and gonads at different developmental phases (2-month-old juveniles, 10-month-old sub-adults, and 3-year-old adults). All isolated tissues were treated for 3 min using liquid nitrogen and then stored at −80 °C.

## 2.2. Total RNA Extraction and cDNA Synthesis of gjSOX9

Total RNA was extracted from the tissues using a UNIQ-10 Column TRIzol Total RNA Isolation Kit II (Sangon Biotech, Shanghai, China), as described by the manufacturer. The total RNA concentration and quality were assessed using a Nanodrop2000 and 1.5% agarose gel electrophoresis, respectively. Reverse transcription was performed with 1.0 μg of total RNA using TransScript One-Step gDNA Removal and cDNA Synthesis SuperMix (TransGen Biotech, Beijing, China) with the Anchored Oligo(dT)18 Primer, according to the manufacturer's protocol.

## 2.3. cDNA Fragment Cloning and Sequence Analyses

Degenerate primers were designed according to the conserved sequence of *G. japonicus* SOX9 (XM_015422341.1, NCBI) and its orthologs in *Eremias multiocellata* (KF700266.1, NCBI) [28], which is genetically close to *G. japonicus*. The gjSOX9 cDNA fragment was amplified via a PCR using the degenerate primers gjSOX9F1/R1, and the PCR products were recombined into the PMD19-T Vector, followed by transformation into *Escherichia coli* DH5$\alpha$ competent cells. Positive clones were identified and sequenced using M13 F/R primers. Sequences were determined using DNASTAR5.01 built-in's SeqMan and confirmed using BLAST analyses (blast.ncbi.nlm.nih.gov/Blast.cgi, accessed on 10 June 2022). The available cDNA fragment was used to design a gene-specific primer (GSP1/2/3/4) for the following 5'/3'-RACE PCR analysis (the rapid amplification of cDNA ends).

## 2.4. Cloning of the Full Length of gjSOX9 cDNA Using RACE Technology

The full length of gjSOX9 was determined using RACE technology, following the manufacturer's instructions (SMARTer$^{TM}$ RACE 5'/3' kit) (TaKaRa, Dalian, China). Briefly, RACE-ready cDNA (first-strand cDNA) was generated based on total RNA, using 5'-CDS primer A and SMARTerII A (provided in the kit). The 5' RACE cDNA was amplified for a first round of PCR using the primer GSP1 (outer primer) and Universal Primer A Mix (UPM) (provided in the kit), followed by a second round of PCR using primer GSP2 (inner primer) and Nested Universal Primer A (NUP) (provided in the kit); 3'-RACE cDNA was amplified using primers GSP3 (outer primer) and UMP for the first round of PCR, and using the primers GSP4 (inner primer) and NUP for the second round of PCR, respectively. PCR product cloning and sequence determination were conducted as described above, and the full-length gjSOX9 cDNA sequence was analyzed using DNASTAR5.01. All primers in this study were designed using Primer Premier5.0 and synthesized by Sangon Biotech (Shanghai, China) (Table 1).

**Table 1.** The primers and their sequences used in this study. UPM represents universal primer A mix and NUP represents nested universal primer A (provided in the kit).

| Primer Name | Sequence | Usage |
|---|---|---|
| *gjSOX9*F1 | TTCCGAGAYGTGGACATTGG | partial sequences acquired |
| *gjSOX9*R1 | GGGCCTYTGGWTGGGRYTCATGT | partial sequences acquired |
| *gjSOX9*-5'GSP1 | CAGATTCACCCCAGTCTTCGTC | 5'RACE |
| *gjSOX9*-5'GSP2 | CCCACATTGACTTCCGAGACG | 5'RACE |
| *gjSOX9*-3'GSP3 | TCCAGTCGTAGCCCTTCAGCACCT | 3'RACE |
| *gjSOX9*-3'GSP4 | CTTGTCCTCGTCGCTCTCCTTCTTC | 3'RACE |
| UPM | CTAATACGACTCACTATAGGGCAAGC AGTGGTATCAACGCAGAGT | 5'/3'RACE |
| NUP | CTAATACGACTCACTATAGGGC | 5'/3'RACE |
| M13F | GTAAAACGACGGCCAGT | clone analysis |
| M13R | CAGGAAACAGCTATGAC | clone analysis |
| *gjSOX9*-RT-F | CGTCTGGATGTGTAAGC | qPCR |
| *gjSOX9*-RT-R | TGATGTGTGTCCTCTGC | qPCR |
| *Actin*-F | CCCCAAAGCCAACAGAGA | qPCR |
| *Actin*-R | ACGCCATCACCAGAGTCCA | qPCR |

## 2.5. gjSOX9 Expression Pattern Analysis

The gjSOX9 mRNA expression was quantified via a reverse-transcription quantitative PCR (RT-qPCR) using the Trans-Start Tip Green qPCR Super Mix kit (TransGen Biotech, Beijing, China). The cDNA template was prepared as described in Section 2.2. The RT-qPCR was amplified in a 20 μL reaction volume including the cDNA template (2.0 μL, 50 ng), gjSOX9-RT-F/R primer (0.2 μM), and 2×TransStart Tip Green qPCR Super Mix (10 μL). The relative expression level was calculated using the $2^{-\Delta\Delta CT}$ method with three repeats for each sample and normalized to the *G. japonicus β-actin* gene (XM_015424816, NCBI) using the PCR primer *Actin*-F/R. Statistical analyses were performed using SPSS 25.0, with a one-way analysis of variance (ANOVA) and individual *t*-tests for the mean $\pm$ standard error ($p < 0.01$). The primers used for the RT-qPCR are listed in Table 1.

## 2.6. Physicochemical Property Analyses and Phylogenetic Reconstruction of gjSOX9

gjSOX9's physicochemical properties were predicted using bioinformatic tools (Table 2). First, the open reading frame (ORF) of the gjSOX9 cDNA was determined (ORF finder), and its encoded amino acid sequence was deduced using the translation program in the Expert Protein Analysis System (Expasy). Next, the theoretical protein molecular weight (MW) and isoelectric point (pI) were analyzed using ProtParam. The transmembrane domain and N-terminal signal peptide were assessed using the TMHMM2.0 (https://services.healthtech.dtu.dk/services/TMHMM-2.0/, accessed on 3 April 2023) and SignalP4.1 online software (https://services.healthtech.dtu.dk/services/SignalP-4.1, accessed on 3 April 2023), respectively. Subcellular localization was determined using Cell-PLoc 2.0. A protein sequence alignment was carried out using DNAMAN9.0. The phylogenetic tree was reconstructed using MEGA11.0, with the neighbor-joining method (Bootstrap 1000) based on amino acid sequence alignment, and was modified using the online software iTol (version 6.7.2) (https://itol.embl.de, accessed on 3 April 2023). The gjSOX9 orthologs in the phylogenetic tree were retrieved from the NCBI database (Table S1).

**Table 2.** Online software or tools used in this study.

| Item Predicted | Software/Database | Web Site |
| --- | --- | --- |
| ORF (open reading frame) | ORF finder | www.ncbi.nlm.nih.gov/projects/gorf (accessed on 21 October 2023) |
| Protein sequence analyses | Translate tool | http://web.expasy.org/translate/ (accessed on 21 October 2023) |
| Physicochemical properties | ProtParam | http://web.expasy.org/protparam (accessed on 21 October 2023) |
| Protein transmembrane | TMHMM2.0 | https://services.healthtech.dtu.dk/services/TMHMM-2.0/ (accessed on 21 October 2023) |
| Hydrophilic analysis | ProtScale | www.expasy.org/cgi-bin/protscale.pl (accessed on 21 October 2023) |
| Signal peptide | SingalP4.1 | https://services.healthtech.dtu.dk/services/SignalP-4.1 (accessed on 21 October 2023) |
| Protein motif | MEME5.5.2 | http://meme-suite.org/meme/tools/meme (accessed on 21 October 2023) |
| Conserved domain | CDD | www.ncbi.nlm.nih.gov/Structure/cdd/cdd.shtml (accessed on 21 October 2023) |
| Subcellular localization | Cell-PLoc 2.0 | www.csbio.sjtu.edu.cn/bioinf/Cell-PLoc-2 (accessed on 21 October 2023) |

## 2.7. Analysis of Conserved Regions of gjSOX9 and the gjSOX Family

Members of the gjSOXs family underwent genome-wide screening and were determined based on bioinformatics tools [19]. Briefly, the protein sequence file of *G. japonicus* was retrieved from the NCBI database (GCF_001447785.1). The Hidden Markov Model (HMM) profile of the conserved HMG-Box motif (Pfam: PF00505) was downloaded from the Pfam database. Protein motifs were analyzed using MEME (version 5.5.2) online software (https://meme-suite.org/meme/tools/meme, accessed on 10 April 2023) [29]. Based on HMMER 3.3 software (Hmmsearch, E $< 1 \times 10^{-10}$) [30], all putative gjSOX members were obtained by screening protein sequences of *G. japonicus*. Next, the gjSOX members

were determined using SMART analyses of the putative protein data after BLASTp analyses and eventually confirmed using CDD tools [31,32]. Putative SOX genes without encoding HMG motifs and redundant genes were excluded.

Protein motifs and domains were predicted using the corresponding online software (Table 2). Multiple sequence alignments of HMGs between gjSOX9 and its orthologs were performed using DNAMAN9.0, and the phylogenetic tree of gjSOX9 and the gjSOX family was reconstructed as described above. Integrative analyses of phylogenetic data and conserved motifs were conducted using iTol software as mentioned above [33].

### 2.8. gjSOX9 Evolution within the gjSOXs Family and Family Orthologs Among Species

To investigate the phylogenetic evolution of gjSOX9 among its family homologs, sequences of the respective gjSOX family orthologs were retrieved from the NCBI database from several species, including *Zootoca vivipara, Xenopus laevis, Trachemys scripta elegans, Pogona vitticeps, Mus musculus*, and *Danio rerio* (Table S2). The phylogenetic tree was reconstructed using the above-mentioned method based on multiple sequence alignment.

## 3. Results

### 3.1. gjSOX9 Sequence and Physiochemical Properties

In this study, the full length of gjSOX9 cDNA was cloned and analyzed, consisting of 1895 bp with the 1482 bp ORF encoding 494 amino acid residues, a 345 bp 5'-terminal untranslated region (UTR), and a 68 bp 3'-UTR (GenBank, OQ935362) (Figure 1). The alignment data indicated only three nucleotide differences out of 1485 bp between the gjSOX9 cDNA cloned in this study (NCBI, OQ935362) and the genome-predicted SOX9 cDNA (NCBI, XM_015422341) in *G. japonicus*, as well as a single amino acid difference out of 494aa between both cDNA deduced amino acid sequences (NCBI, WGV33816.1/XP_015277827.1) (Figure S1), suggesting the reliability of the gjSOX9 cDNA sequence cloned in this study.

Bioinformatics analyses indicated that gjSOX9, a hydrophilic nucleoprotein with a theoretical MW of 55.29 kDa and a pI of 6.4, has a characteristic HMG-Box DNA binding domain with 75 aa at sites 104–178 and lacks a hydrophobic transmembrane region and signal peptide cleavage site (Table 3; Figure S2), which suppose that gjSOX9 is not transmembrane or secretory protein.

**Table 3.** Comparison with gjSOX9 and its gjSOXs family members. HI* represents hydrophilicity index.

| SOX Family | Accession No. | Size (aa) | MW (kDa) | pI | HI* | Subcellular Localization |
|---|---|---|---|---|---|---|
| gjSOX2 | XP_015262440.1 | 384 | 40.74 | 9.92 | −0.57 | Cytoplasm; nucleus |
| gjSOX3 | XP_015267683.1 | 318 | 34.01 | 9.62 | −0.61 | Cytoplasm; nucleus |
| gjSOX4 | XP_015271872.1 | 420 | 44.34 | 6.74 | −0.67 | Nucleus |
| gjSOX5 | XP_015268168.1 | 764 | 84.30 | 6.12 | −0.77 | Nucleus |
| gjSOX6 | XP_015277789.1 | 787 | 87.40 | 6.54 | −0.82 | Nucleus |
| gjSOX7 | XP_015284766.1 | 375 | 41.46 | 6.16 | −0.68 | Nucleus |
| gjSOX8 | XP_015279835.1 | 325 | 35.64 | 6.28 | −1.02 | Cytoplasm; nucleus |
| gjSOX9 | WGV33816.1 | 494 | 55.35 | 6.31 | −1.09 | Nucleus |
| gjSOX10 | XP_015261463.1 | 461 | 49.89 | 6.20 | −0.84 | Nucleus |
| gjSOX11 | XP_015262936.1 | 417 | 45.62 | 5.09 | −0.85 | Cytoplasm; nucleus |
| gjSOX12 | XP_015279104.1 | 291 | 32.88 | 8.82 | −0.87 | Cytoplasm; nucleus |
| gjSOX13 | XP_015262805.1 | 615 | 69.24 | 6.31 | −0.87 | Nucleus |
| gjSOX14 | XP_015276981.1 | 240 | 26.64 | 9.68 | −0.61 | Cytoplasm; nucleus |
| gjSOX17 | XP_015282499.1 | 177 | 18.63 | 5.94 | −0.55 | Nucleus |
| gjSOX18L | XP_015284209.1 | 398 | 44.32 | 8.66 | −0.76 | Nucleus |
| gjSOX30 | XP_015272912.1 | 778 | 85.68 | 5.75 | −0.76 | Nucleus |

```
   1    ACATGGGGATTCGACTTCTTTGCCCTCCCCGCCAAACTTCTCTTTGACCGTCCGCGAACG   60
  61    GATCAGCTGCCCACATGTCTGTTTCTTGAGAGAGGAGAAAACTGCAGCGACAACTTTGCA  120
 121    AGGCGCCTCTGATCCGCGTTTCTCTCTCTCTCTYTCTTTCAAGTTTCTGAGAACCAGGTC  180
 181    AATCCCGTTTGCGAACTTTTGGGTGGTCCTGCTCGTTTTTAAATGCTGTTTTAGGAGACC  240
 241    CTGCTCCTGCTGATTTCCAATCCCTCCCCACCCTCCTCTTCCCTCGCCCCTCTCTCTCCT  300
 301    CCATTTTCATCGCCCCCCCCCATCTTTGCTTTGCCGCTTTCTCGCatgaatctcctcgac  360
   1                                                       M  N  L  L  D     5
 361    cccttcatgaagatgacagaagagcaggagaaatgtctgtccggcgcccccagccccacc  420
   6       P  F  M  K  M  T  E  E  Q  E  K  C  L  S  G  A  P  S  P  T    25
 421    atgtcggacgactccgccggctcgccttgcccttcgggctccggatcggacaccgagaac  480
  26       M  S  D  D  S  A  G  S  P  C  P  S  G  S  G  S  D  T  E  N    45
 481    acccgaccgcaggaaaacaccttccccaagaacgacccggacttgaagaaggagagcgac  540
  46       T  R  P  Q  E  N  T  F  P  K  N  D  P  D  L  K  K  E  S  D    65
 541    gaggacaagttcccggtgtgcatccgcgaggccgtgagccaggtgctgaagggctacgac  600
  66       E  D  K  F  P  V  C  I  R  E  A  V  S  Q  V  L  K  G  Y  D    85
 601    tggacgctggtgcccatgccggtgcgggtgaacggctccagcaagaacaagcccacgtc  660
  86       W  T  L  V  P  M  P  V  R  V  N  G  S  S  K  N  K  P  H  V   105
 661    aagcggcccatgaacgccttcatggtctgggcgcaggcggcccgcaggaagctggccgac  720
 106       K  R  P  M  N  A  F  M  V  W  A  Q  A  A  R  R  K  L  A  D   125
 721    cagtaccccacctgcacaacgccgagctcagcaagaccctgggcaaactctggaggtta  780
 126       Q  Y  P  H  L  H  N  A  E  L  S  K  T  L  G  K  L  W  R  L   145
 781    ctgaatgagagtgagaaacgtccatttgtggaggaggctgagaggcttagggtgcagcac  840
 146       L  N  E  S  E  K  R  P  F  V  E  E  A  E  R  L  R  V  Q  H   165
 841    aaaaaagaccatcccgactataagtaccagccccggagaagaaaatcagtcaagaatggg  900
 166       K  K  D  H  P  D  Y  K  Y  Q  P  R  R  R  K  S  V  K  N  G   185
 901    caggctgagcaggaggaagggtctgagcaaacccacatctctccgaacgccatcttcaag  960
 186       Q  A  E  Q  E  E  G  S  E  Q  T  H  I  S  P  N  A  I  F  K   205
 961    gccttgcaggcagattcaccccagtcttcgtcgagcatgagtgaagtgcactcccctggg 1020
 206       A  L  Q  A  D  S  P  Q  S  S  S  S  M  S  E  V  H  S  P  G   225
1021    gagcattccggccaatctcaagggccacccacccctcctacaacccctaaaacagatgtc 1080
 226       E  H  S  G  Q  S  Q  G  P  P  T  P  P  T  T  P  K  T  D  V   245
1081    caacctggaaagcaggacctgaagcgagaaggacgcccctgccagaaggagggaggcag 1140
 246       Q  P  G  K  Q  D  L  K  R  E  G  R  P  L  E  G  G  R  Q   265
1141    ccgccccacattgacttccgagacgtggacattggggagctcagcagtgatgtcatctcc 1200
 266       P  P  H  I  D  F  R  D  V  D  I  G  E  L  S  S  D  V  I  S   285
1201    aacattgagacctttgatgtcaatgagtttgaccagtatctcccacccaatggccaccca 1260
 286       N  I  E  T  F  D  V  N  E  F  D  Q  Y  L  P  P  N  G  H  P   305
1261    ggtgttccagtcacccatggccaacctggccaagtcacttatactggcagctatggaatc 1320
 306       G  V  P  V  T  H  G  Q  P  G  Q  V  T  Y  T  G  S  Y  G  I   325
1321    agcagcacaacagccacgccagcaggtactggtcacgtctggatgtctaagcaacaacca 1380
 326       S  S  T  T  A  T  P  A  G  T  G  H  V  W  M  S  K  Q  Q  P   345
1381    ccatctcagcagccaccgtcccaagctcagcagcaagcatcacagcaacagcagcataca 1440
 346       P  S  Q  Q  P  P  S  Q  A  Q  Q  Q  A  S  Q  Q  Q  H  T   365
1441    ctaaccaccctgagtagtgaacaggggcaacctcagcagaggacacacatcaaaactgag 1500
 366       L  T  T  L  S  S  E  Q  G  Q  P  Q  Q  R  T  H  I  K  T  E   385
1501    cagctcagtcccagccattacactgagcagcagcagcattctcctcagcagatcagctac 1560
 386       Q  L  S  P  S  H  Y  T  E  Q  Q  Q  H  S  P  Q  Q  I  S  Y   405
1561    acttccttcaacctccagcactacagttcctcctacccaactatcactcgttcccagtat 1620
 406       T  S  F  N  L  Q  H  Y  S  S  S  Y  P  T  I  T  R  S  Q  Y   425
1621    gactacacagaccaccagaactccaactcctactacagccatgcagccagccagagttcc 1680
 426       D  Y  T  D  H  Q  N  S  N  S  Y  Y  S  H  A  A  S  Q  S  S   445
1681    agtctctattcaacattcacctacatgaacccccacccagaggccaatgtacacaccgatt 1740
 446       S  L  Y  S  T  F  T  Y  M  N  P  T  Q  R  P  M  Y  T  P  I   465
1741    gcaggtactacaggggtgccttccattcctcagacccacagcccacaacactgggaacag 1800
 466       A  G  T  T  G  V  P  S  I  P  Q  T  H  S  P  Q  H  W  E  Q   485
1801    cctgtctacacacaactcacaaggccataaGGCTCTGAAAGATGGCCAAACGTCCTCAGA 1860
 486       P  V  Y  T  Q  L  T  R  P  *                                494
1861    CTTACAAAAAAAAAAAAAAAAAAAAAAAAAAAAAAAAAA                       1890
```

**Figure 1.** Nucleotide sequence and deduced amino acid sequence of gjSOX9 cDNA. The red "letters" represent the start codon "atg" and the end codon "taa". The yellow highlighted area is the HMG domain (site104–178aa). Upstream of the start codon is 5′UTR, and downstream of the end codon is 3′UTR. * means the absence of amino acids in this position.

### 3.2. gjSOX9 Expression Patterns in G. japonicus

In this study, gjSOX9 expression was detected in all six adult tissues (Figure 2). In the gonads, the gjSOX9 expression amount in testes was approximately 15 times higher than in ovaries. In the liver, the gjSOX expression amount in females was significantly higher than in males. In the kidneys, gjSOX9 had almost the same expression levels in males and females, suggesting equally important roles in kidney tissue metabolism and function maintenance in both sexes [17]. In the brain, gjSOX9 exhibited the highest expression levels in females and the second highest expression levels in males among all six adult tissues, indicating significant roles of gjSOX9 in the functional maintenance of neural stem cells [16]. In addition, very low gjSOX9 expression levels indicated insignificant roles of gjSOX9 in the heart and muscle, regardless of sex.

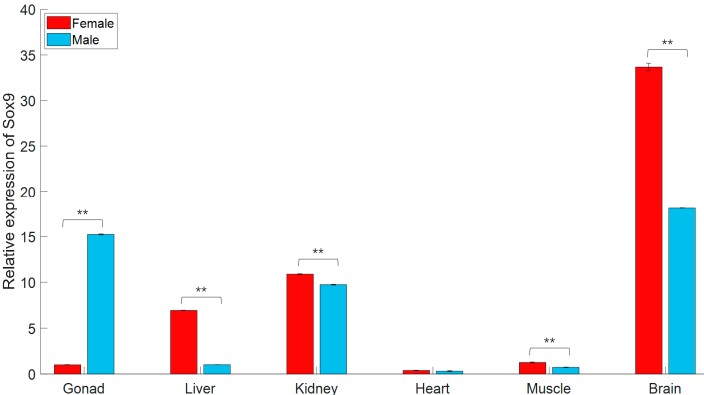

**Figure 2.** gjSOX9 expression in various adult tissues in female and male individuals. The relative expression levels were analyzed using an RT-qPCR and calculated using the $2^{-\Delta\Delta CT}$ method. *G. japonicus β-actin* was used as the reference gene. All data are represented as the mean $\pm$ SE ($n = 3$ independent experiments) (** $p < 0.01$).

Distinct differences in the expression of gjSOX9 in gonad tissues appeared between the non-reproductive and reproductive phases. As shown in Figure 3A, whether in the non-reproductive or reproductive phase, gjSOX9 expression in testes was distinctly higher than in ovaries. Comparatively, gjSOX9 expression levels in testes or ovaries in non-reproductive phase decreased a lot than those in reproductive phase.

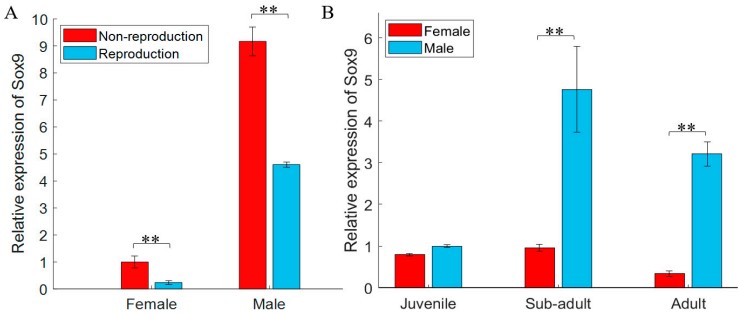

**Figure 3.** gjSOX9 expression analyses in gonad tissues. (**A**) gjSOX9 expression in adult gonad tissues during non-reproductive (November) and reproductive stages (May); (**B**) gjSOX9 expression at different development stages (Juvenile for 2-month-old individuals, Sub-adult for 10-month-old individuals, Adult for 3-year-old individuals). The relative expression levels were analyzed using a RT-qPCR and calculated using the $2^{-\Delta\Delta CT}$ method. *G. japonicus β-actin* was used as the reference gene. All data are represented as the mean $\pm$ SE ($n = 3$ independent experiments) (** $p < 0.01$).

The special spatiotemporal expression of gjSOX9 existed in gonad tissues at different developmental phases (2-month-old juveniles, 10-month-old sub-adults, and 3-year-old adults). Figure 3B indicates that in the 2-month-old juveniles, gjSOX9 expression was

kept considerably low in testes and ovaries. But in sub-adults, gjSOX9 expression sharply increased in testes but still maintained a similarly low level of expression as that in juvenile ovaries. In adults, gjSOX9 expression decreased significantly in testes and in ovaries.

### 3.3. gjSOX9 Phylogenetic Analyses

Phylogenetic analyses of gjSOX9 revealed that 20 SOX9 orthologs were divided into two clades (Figure S3): one large clade consisted of 17 SOX9 proteins, including gjSOX9, and the other clade consisted of three SOX proteins from fishes (*Danio rerio* SOX9a/9b, drSOX9a/drSOX9b; *Orizias latipes* SOX9, olSOX9). Within the large clade, gjSOX9 shared the closest genetic relationships with its homologies from gecko (*Eublepharis macularius* SOX9, emSOX9; *Sphacrodactylus townsendi* SOX9, stSOX9), lizard (*Hemicordylus capensis* SOX9, hcSOX9; *Anolis carolinensis* SOX9, acSOX9), and snake (*Python bivittatus* SOX9, pbSOX9). Four SOX9 orthologs from mammals clustered together, including duckbill (*Ornithorhynchus anatinus* SOX9, oaSOX9), dog (*Canis lupus familiaris* SOX9, clSOX9), human (*Homo sapiens* SOX9, hsSOX9), and mouse (*Mus musculus* SOX9, mmSOX9). In addition, three SOX9 orthologs from zebra finch (*Taeniopygia guttata* SOX9, tgSOX9), chicken (*Gallus gallus* SOX9, ggSOX9), and turkey (*Meleagris gallopavo* SOX9, mgSOX9) also clustered closely.

### 3.4. Comparison with gjSOX9 and gjSOX Family

Based on HMMER, BLASTp analyses, and CDD confirmation, 16 gjSOX proteins were eventually identified as gjSOX family members in *G. japonicus*, including gjSOX2–14, gjSOX17, gjSOX18L, and gjSOX30 (Table 3). Comparative data indicated that the full length of gjSOX9 (494aa) was slightly longer than the average size (452aa) of the 16 gjSOX families, whose sizes ranged from 177aa (gjSOX17) to 787aa (gjSOX6). Of the identified gjSOX members, gjSOX9 shared highly similar physicochemical properties and a subcellular location with gjSOX10 (Table 3) and almost all hydrophilic properties with the other 15 members. Cellular localization data showed that 10 gjSOX members were present in the nucleus, while the other 6 members were present in both the cytoplasm and nucleus, suggesting conserved region sequences and functional differences among the 16 gjSOX members (Table 3).

### 3.5. Motif and Domain Analyses of gjSOX9 and gjSOX Family

Motifs, as super-secondary structures of proteins, possess several highly conserved amino acid residues and perform the particular functions of the protein, which provide basic knowledge for the prediction of protein function. The data reveal that the gjSOXs family contains 5 types of motifs (motif1–8) (Figure 4) and gjSOX9 contains 4 motifs (motif1,2,4,7), while the other 15 gjSOXs members contain 1–5 motifs. Of the eight motifs identified, two (motif1 and motif2) were tightly aligned and were located in various regions of the gjSOX family (except for gjSOX8 and gjSOX17). Additionally, each motif exhibited a different degree of sequence conservation (Figure S4).

Domains consist of one or more motifs and have an independent, stable existence in the protein structure. The data reveal that sixteen gjSOX members contain 12 different conserved domains, with each member having 1–2 domains. gjSOX9 harbored two typical domains: high-mobility group box (HMG) at site 104–178aa and SOX_N terminal at site 22–94aa. gjSOX9 shared highly consistent conserved domains with gjSOX10 and different conversed domains with other groups of members such as gjSOX2 and gjSOX3 (SOXB) (Figure 5).

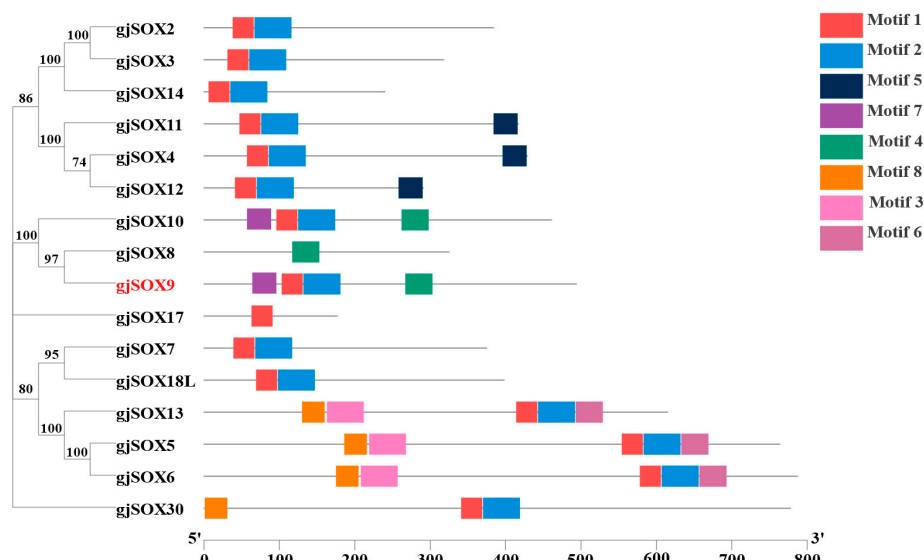

**Figure 4.** Phylogenetic relationships and conserved motif composition of gjSOX. The left part represents the phylogenetic analyses and the right part indicates the motif compositions of 16 gjSOX proteins. The phylogenetic tree was reconstructed using MEGA11.0 software, using the neighbor-joining method with 1000 bootstrap replicates based on amino acid sequence alignment. MEME was used to predict motifs, which are represented by different colored boxes numbered 1–8. Phylogenetic analyses and motif data were integrated using the software TBtools [34]. The gjSOX9 ortholog sequences in the phylogenetic tree were retrieved from the NCBI database (Table S1).

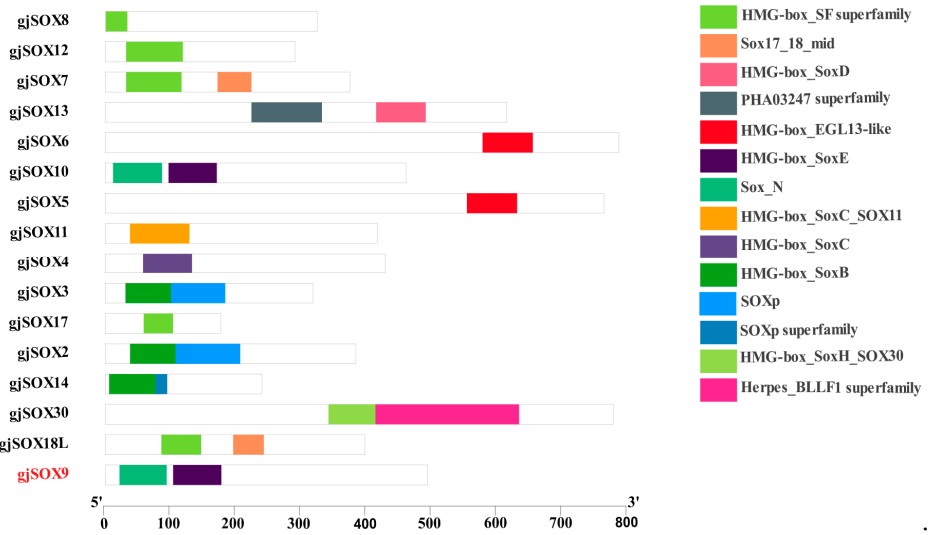

**Figure 5.** Conserved domain compositions of 16 gjSOXs members. The 14 conserved domains were identified using the Batch CD-Search tool in the NCBI database and are represented using differently colored boxes.

The multiple sequence alignments indicate that amino acid variations occur in the highly conserved HMG-Box region among the 16 gjSOX members (Figure S5), and gjSOX9 shares the unique sequence "YKYQPRRR" with gjSOX8 and gjSOX10. The phylogenetic analyses divided the 16 gjSOX family members into five groups, including gjSOX8,9,10 (SOXE), gjSOX2,3,14 (SOXB), gjSOX4,11,12 (SOXC), gjSOX5,6,13 (SOXD), and gjSOX7,18L (SOXF), as well as gjSOX17 (SOXF) and gjSOX30 (SOXH), isolated alone (Figure 4).

### 3.6. Evolution Analyses of gjSOX9 and Its Family Members among Species

In this study, 150 SOX homologs were utilized for phylogenetic analysis, including 16 gjSOX members and their respective orthologs from the lizard *Zootoca vivipara*, frog *Xenopus laevis*, turtle *Trachemys scripta elegans*, lizard *Pogona vitticeps*, mouse *Mus musculus*, and zebrafish *Danio rerio* (Table S2). The available data revealed that the 16 gjSOX members were divided into six groups: SOXB1 (gjSOX2,3), SOXB2 (gjSOX14), SOXC (gjSOX4,11,12), SOXD (gjSOX5,6,13), SOXE (gjSOX8,9,10), SOXF (gjSOX7,17,18L), and SOXH (gjSOX30) (Figure 6). Within the SOXE group, gjSOX9 shared the closest genetic relationships with *Pogona vitticeps* SOX9 (pvSOX9), *Zootoca vivipara* SOX9 (zvSOX9), *Trachemys scripta elegans* SOX9 (tsSOX9), *Mus musculus* SOX9 (mmSOX9), *Xenopus laevis* SOX9L/S (xlSOX9L/S), and *Danio rerio* SOX9a/b (drSOX9a/b), and close relationships with gjSOX8 and gjSOX10, as well as their respective orthologs.

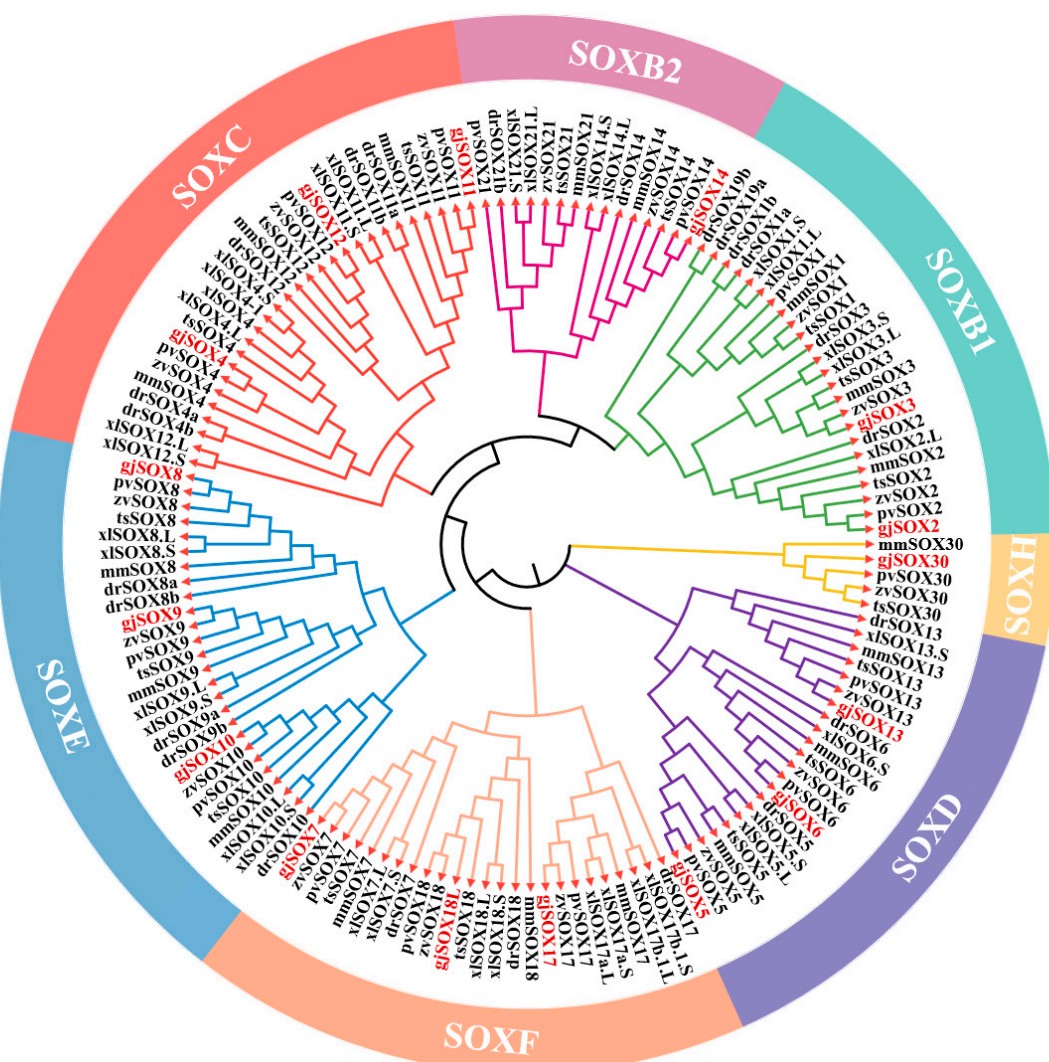

**Figure 6.** Phylogenetic analyses of gjSOX9 and its family members. The phylogenetic tree was reconstructed using MEGA11.0 software, using the neighbor-joining method with 1000 bootstrap replicates based on amino acid sequence alignment. The gjSOX9 family member ortholog sequences in the phylogenetic tree were retrieved from the NCBI database (Table S2). gjSOX9 and its family members are highlighted in red letters; zv for *Zootoca vivipara*, xl for *Xenopus laevis*, ts for *Trachemys scripta elegans*, pv for *Pogona vitticeps*, mm for *Mus musculus*, and dr for *Danio rerio*.

## 4. Discussion

### 4.1. gjSOX9 cDNAs Sequence Analyses

Currently, several SOX9 cDNAs are cloned, and their amino acids sequences were deduced according to cloned cDNA, including the mud crab *Scylla paramamosain* SOX9 (spSOX9,491aa) [21], *Alligator sinensis* SOX9 (asSOX9, 494aa) [17], lambari fish *Astyanax altiparanae* SOX9 (aaSOX9, 483aa) [22], and orange-spotted grouper *Epinephelus coioides* SOX9 (ecSOX9,479aa) [35]. In this study, gjSOX9 cDNA was found to encode 494aa, which shares a completely identical size with asSOX9 from a reptile species. The available data indicated the sizes of 20 SOX9 orthologs in this study ranged from 365aa (*Python bivittatus* SOX9, pbSOX9) to 526aa (*Anolis carolinensis* SOX9, acSOX9), with an average of 481aa (Table S1), which was slightly shorter than that of gjSOX9 (494aa). Seemly, there were no direct relationships between SOX9 size and species evolution.

As master transcription factors, all SOX9 orthologs in this study shared highly similar hydrophilicity indices and subcellular locations (Table S1). Phylogenetic analyses revealed that gjSOX9 had a close genetic relationship with gecko emSOX9 and stSOX9. Totally, gjSOX9 orthologs from close genetic species clustered tightly together.

### 4.2. gjSOX9 Expression Patterns in G. japonicus

In this study, gjSOX9 exhibited bisexual, male-biased, and typical spatiotemporal expression patterns in gonad tissues (Figure 3), which was highly consistent with SOX9 expression in the mud crab *Scylla paramamosain* and in the caudate amphibian *Pleurodeles waltl* [21,36]. This suggested that gjSOX9, as a master transcription factor, mainly regulates the expression of male-related genes involved in sex determination, testis development, maturation, and male fertility maintenance. In olive flounder *Paralichthys olivaceus*, the expression levels of poSOX9a and poSOX9b were both higher in male individuals than female individuals during sex differentiation, whereas the expression levels of poSOX9a and poSOX9b were both higher in the testis than in the ovary in adult gonads, implying that two SOX9 paralogs play different roles in sex differentiation, spermatogenesis, and gonadal function maintenance [11].

This study confirmed gjSOX9 expression in the liver, kidney, heart, muscle, and brain tissues in addition to gonad tissues (Figure 2). Particularly high expression levels were observed in the brain, kidney, and female liver tissues, implying crucial roles of gjSOX9 in these tissues during tissues development and function maintenance. Previous studies reveal a high level of expression of SOX9 during nephrogenesis in *Alligator sinensis* [17] and in the eyestalk and cerebral ganglion tissues of the mud crab *Scylla paramamosain* [21], suggesting that SOX9 also plays substantive roles in the development of these tissues. In zebrafish, SOX9a shares distinct expression patterns with SOX9b but overlaps in some regions of the brain, head skeleton, and fins during embryogenesis. Differently, for adult individuals, SOX9a is expressed in the brain, muscle, fin, and testis, whereas SOX9b is only expressed in ovarian tissues, indicative of the unique functions of the two SOX9 paralogs in different development phases [8]. But in olive flounder, SOX9a and SOX9b expression is also found in several somatic tissues such as the heart, liver, kidney, muscle, brain, and gill in addition to the testis gonad tissue. Seemly, the expression differences in SOX9a and SOX9b exist in olive flounder and zebrafish species [8,11].

### 4.3. Variation in Conserved Region in gjSOX9 and Its Family Members

For the 16 gjSOX members, amino acid variations were observed in their highly conserved HMG-Box regions (Figure S5). gjSOX9 shares the unique sequence "YKYQPRRR" with gjSOX8,10, which is distinct from its counterparts in other gjSOX members such as "YKYRPRRR" in gjSOX2–4 or "QQQEQIAR(K)" in gjSOX5,6,13. Moreover, the sequence "YKYQPRRR" was first observed in all 20 gjSOX9 orthologs and all 26 SOXE members (SOX8–10) among the species in this study, which implies that sequence (YKYRPRRR) is unique and functionally important in SOXE group members. Amino acid variation in the conserved HMG-Box domain means that the gjSOX family can recognize and bind diverse

DNA regulatory regions, implying that the ancestor gjSOX gene underwent divergent evolution to generate functional differences during long-term species evolution.

### 4.4. Evolutionary Analyses of SOX9 and Its Family Members among Species

SOX9 and its family are found in almost all animals, including invertebrates and vertebrates [4,37]. SOX9 and its group members (SOX8 and 10) (SOXE) are mainly implicated in the reproductive system, and other group members, such as SOXB members, are mainly involved in the central nervous system [3,7]. Previous studies have shown that different species possess various groups and numbers of SOX members, such as 20 SOX members in eight groups in humans and mice [38] and 24 SOX members in seven groups in *Fugu rubripes* [39]. In this study, 16 gjSOX members in the groups B(B1/2)/C/D/E/F/H were determined in *G. japonicus* (Table 3; Figure 6), and the SOX family grouping in this study was in high agreement with those previously proposed based on HMG identities [1,3,4]. Notably, SOX members in the groups A/G/I/J/K failed to be observed in *G. japonicus*. Possibly, these group members were functionally attenuated and gradually eliminated due to selection pressure during long-term species evolution. SOX family members often appear in the form of a single copy (such as SOX9) in the vast majority of species, but two copies of SOX paralogs (such as SOX9a and SOX9b) often occur in teleost species [3,8,11,12,39]. In the species *Danio rerio*, two SOX9 paralogs (drSOX9a and drSOX9b) are observed [8], and drSOX9b shares the closest genetic relationship with *Orizias latipes* SOX9 (olSOX9) and the second closest genetic relationship with drSOX9a (Figure S3), implying an evolutional difference for drSOX9a and drSOX9b (Figure S3). Generally, for two copies of SOX paralogs, one copy is used for its original function (such as drSOX9a) while the other evolves a new function (such as drSOX9b), suggesting SOX family duplication and functional divergence (neofunctionalization) in these species [8,12,40,41]. The presence of SOX paralogs greatly increases the complexity of gene regulatory landscapes and functional diversities [41,42].

### 4.5. Future Investigations of gjSOX9 in G. japonicus

For a long time, researchers have been eager to discover the underlying complex mechanism of sex determination in *G. japonicus*, which is precisely co-regulated by genetic and environmental factors. A functional analysis of sex-determination (or sex-associated) genes such as *SOX9* and *DMRT1* would provide much valuable knowledge. Therefore, future studies should focus on the molecular regulation of gjSOX9 and the function characterization of other sex-associated genes in *G. japonicus*. Hopefully, newly developed biotechnologies can greatly advance this field: (1) Omics analyses could help identify sex-associated genes and explore their regulation mechanism at various levels. (2) Genome-wide association studies (GWASs) could establish the relationships between special traits (such as sex) and genes (or SNPs) through re-sequence analyses [43,44]. (3) The CRISPR system can contribute to gene function characterization through gene gain (or loss) of function [45].

### 5. Conclusions

In this study, the full-length gjSOX9 cDNA sequence, gjSOX9 expression patterns, and phylogenetic evolution were firstly determined in *G. japonicus*. Moreover, comparisons were made between gjSOX9 and gjSOX family members, as well as family homologs among species. In particular, the conserved sequence (YKYRPRRR) was first observed only in gjSOX9 and the SOXE group (SOX8, SOX9 and SOX10). In short, this study provides valuable insights into the characterization of gjSOX9 and gjSOX family members. Moreover, the available data in this study also help to understand the origin and evolution of SOX9 homologs or even sex-determination mode in reptiles.

**Supplementary Materials:** The supporting information can be downloaded at https://www.mdpi.com/article/10.3390/cimb45110584/s1, Figure S1: Sequence alignment of gjSOX9; Figure S2: Hydrophilic analysis (left) and signal peptide prediction (right) of gjSOX9; Figure S3: Phylogenetic

analyses of gjSOX9 with its homologies; Figure S4: Sequence conservation analyses of motif1–8 among 16 gjSOX members; Figure S5: Multiple sequence alignment of HMGs for 16 gjSOX members; Table S1: gjSOX9 and its orthologs properties; Table S2: gjSOX family members and their respective orthologs in this study.

**Author Contributions:** Conceptualization, Y.Z.; methodology, X.W.; investigation, X.H., R.Z., C.F. and Z.X.; data curation, R.Z. and S.L.; formal analysis, X.H., R.Z., L.X. and S.L.; resources, Z.X. and Y.Z.; project administration, L.X.; funding acquisition, Y.Z. All authors have read and agreed to the published version of the manuscript.

**Funding:** This study was supported by the National Natural Science Foundation of China (No. 31971419; 31170376).

**Institutional Review Board Statement:** The study was conducted in accordance with the Declaration of China and approved by the Institutional Review Board of Wenzhou University (code: wzu-2022-016; date: 5 January 2022).

**Informed Consent Statement:** Not applicable.

**Data Availability Statement:** Data are contained within the article and supplementary materials.

**Conflicts of Interest:** The authors declare no conflict of interest.

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
