# Peer review of "gjSOX9 Cloning, Expression, and Comparison with gjSOXs Family Members in Gekko japonicus"

_cimb, doi:10.3390/cimb45110584_

Round 1
Reviewer 1 Report
Comments and Suggestions for Authors
Dear Editor,
Thank you for inviting me to review the work of Huang et al., which is an example of a nice combination of laboratory work and bioinformatic analyses. The overall scientific significance is good and the work would be interesting for the scientific community.
The Introduction section is well written and targeted to the goal of the study. The Materials and Methods section is written in a reproducible manner. The Results section unfortunately contains some elements of discussion which must be transferred to the Discussion section. I have no remarks concerning the Discussion section. Many redundancies from the Results and the Discussion sections are present within the Conclusion section.
After reading the entire article, I have the following remarks:
MAJOR REMARKS:
The overall significance/merit/novelty of the study should be clearly indicated at the end of the abstract;
Despite not being a native English speaker, some slight language polishing is needed at my opinion because some phrases sound too wordy;
In subsection 3.2. should be briefly explained the units used for measuring of the relative expression;
The conclusion section should be significantly shortened containing only the major conclusions of the study.
MINOR REMARKS:
Line 20: the expression “whether in (non)-reproduction” is unclear and should be changed;
Line 132: I would recommend using the abbreviation “RT-qPCR” instead of the used because reverse-transcription is a subtype of the qPCR;
Line 184: Species name in italics;
Lines 205-211, 214-220, 249-250, 294-307, 323-325: Discussion, not results
Line 354: Genus name with a capital letter.
Based on my remarks I recommend the article for publication after undergoing a revision of the manuscript.
Author Response
Dear my Reviewer,
Thank you very much for your positive and constructive suggestions on our manuscript (cimb-2703883). We have carefully read through the comments and made proper revision. All revisions are colored marked in the revised manuscript. The point-by-point responses and revision details are listed below.
We sincerely appreciate your time and efforts, valuable suggestions and critical comments on our consideration!
Best regards,
Corresponding author

Reviewer 2 Report
Comments and Suggestions for Authors
In the present manuscript, Huang et al. cloned the cDNA sequence of the SOX9 transcription factor from Gekko japonicus. Its relative mRNA expression levels in multiple organs from male and female animals were compared. Its relative mRNA expression levels were also compared between male and female gonad tissues at different developmental and reproductive stages. Phylogenetic and domain conservation analyses were also performed.
The manuscript is clearly written and well-organized. It contributes resources for future studies on the SOX9 transcription factor during Gekko japonicus developmental and reproductive stages.
Minor remarks:
1. In line 114, please consider changing "The full length of gjSOX9 cDNA cloning using RACE technology" to "Cloning of full length gjSOX9 cDNA using RACE technology".
2. In line 130, please consider changing "gjSOX9 expression patterns analyses" to "gjSOX9 expression pattern analysis".
3. In line 133, it reads: "The cDNA template was prepared as described previously". Please include a reference or mention in which section of the manuscript this information can be found.
4. In line 155, please consider changing "Conserved regions analyses of gjSOX9 and gjSOX family" to "Analysis of conserved regions of gjSOX9 and the gjSOX family".
5. In the legends of figures 2 and 3, please include asterisks (**) near the corresponding p-value.
6. In line 265, please consider changing "Motifs and domains analyses of gjSOX9 and gjSOX family " to "Motif and domain analyses of gjSOX9 and gjSOX family".
7. In line 273, please correct "conversation" to "conservation".
Comments on the Quality of English LanguageA few minor issues in the quality of English language. Please see "Comments and Suggestions for Authors
".
Author Response
Dear my Reviewer,
Thank you very much for your positive and constructive suggestions on our manuscript (cimb-2703883). We have carefully read through the comments and made proper revision. All revisions are colored marked in the revised manuscript. The point-by-point responses and revision details are attached.
We sincerely appreciate your time and efforts, valuable suggestions and critical comments on our consideration!
Best regards,
Corresponding authors
